# Nonstationary Dual Averaging and Online Fair Allocation

**Luofeng Liao, Yuan Gao, Christian Kroer**
IEOR, Columbia University
`{ll3530,yg254,ck294}@columbia.edu`

## Abstract

We consider the problem of fairly allocating sequentially arriving items to a set of individuals. For this problem, the recently-introduced PACE algorithm leverages the dual averaging algorithm to approximate competitive equilibria and thus generate online fair allocations. PACE is simple, distributed, and parameter-free, making it appealing for practical use in large-scale systems. However, current performance guarantees for PACE require i.i.d. item arrivals. Since real-world data is rarely i.i.d., or even stationary, we study the performance of PACE on nonstationary data. We start by developing new convergence results for the general dual averaging algorithm under three nonstationary input models: adversarially-corrupted stochastic input, ergodic input, and block-independent (including periodic) input. Our results show convergence of dual averaging up to errors caused by nonstationarity of the data, and recover the classical bounds when the input data is i.i.d. Using these results, we show that the PACE algorithm for online fair allocation simultaneously achieves "best of many worlds" guarantees against any of these nonstationary input models as well as against i.i.d. input. Finally, numerical experiments show strong empirical performance of PACE against nonstationary inputs.

## 1 Introduction

In fair division, the goal is to allocate a set of items, typically assumed divisible, among a set of agents with heterogeneous preferences, while guaranteeing fairness and efficiency properties. In this paper we are interested in how to fairly and efficiency allocate items that arrive *online*: at every time step one item arrives, and we must irrevocably assign it to some agent. Recently, there has been a growing literature on such online fair allocation problems (Azar et al., 2016; Balseiro et al., 2020; Gao et al., 2021; Bateni et al., 2021; Sinclair et al., 2021; Banerjee et al., 2022). Real-world systems that can be captured by such settings include Internet advertising systems, job recommender systems, cloud computing platforms, and many more. One of the key challenges in such problems is to balance the (often conflicting) goals of overall efficient resource utilization with fairness guarantees for the individual agents.

For this setting, Gao et al. (2021) shows that a simple mechanism called PACE (Pace According to Current Estimated utility) generates asymptotically fair and efficient allocations when the item arrivals are drawn in an i.i.d. manner. PACE gives each agent a per-time-step budget of faux currency, and the fair allocation is achieved by having agents participate in first-price auctions for each item, using the faux money. By guaranteeing that each agent asymptotically spends their budget at the correct rate, the resulting allocations and prices converge to what is known as a *competitive equilibrium from equal incomes* (CEEI), which guarantees both fairness and efficiency. In PACE, each agent maintains a *pacing multiplier* to control their spending over time, and the pacing multipliers are updated based on buyers' budgets and cumulative utilities. This is similar to how budget-management

36th Conference on Neural Information Processing Systems (NeurIPS 2022).

systems work in Internet ad auctions. PACE is highly decentralized due to its auction-based allocation, it does not require dividing the item, and it is also completely parameter free. This makes it suitable for large-scale practical implementation.

Yet in many large-scale settings, such as the context of fair recommender systems (Kroer et al., 2021; Kroer and Stier-Moses, 2022) or Internet advertising, we would not expect items to be drawn i.i.d. from a single distribution. One alternative is to assume that data arrives adversarially. However, this leads to very pessimistic negative results and is not an accurate representation of the data one would expect to see in practice. Instead, one would expect the data to have a strong stochastic component, but with changes over time, e.g., due to flow of traffic, breaking news events, or system updates (Esfandiari et al., 2018; Balseiro et al., 2020).

Motivated by the above considerations, we study online fair allocation when the data exhibits nonstationary behavior. In particular, we focus on the performance of the PACE algorithm of Gao et al. (2021). We ask

*How does PACE behave when nonstationarity is present in the stream of items?*

We show that, under several data-input models, the fairness and efficiency guarantees of the PACE algorithm are still preserved, up to errors due to the nonstationarity of the data input. In this sense, we significantly extend the main results in Gao et al. (2021). To show these results, we first consider the more general setting of nonstationary stochastic optimization and develop new performance guarantees for dual averaging in this setting. Given the ubiquitous use of dual averaging in online and stochastic optimization, our results are of broader interest beyond (fair) resource allocation.

## 1.1 Summary of Contributions

**Novel convergence results for dual averaging under three nonstationary settings.**

We analyze the dual averaging (DA) algorithm for nonstationary stochastic optimization under different data input models, namely, (1) mildly corrupted, (2) ergodic and (3) periodic input data. Specifically, we consider the composite dual averaging algorithm, where the composite term is strongly convex. We show that, in all cases, the iterates generated by dual averaging (DA) converge to the optimal solution in mean square, where the bound on the mean-square error decomposes into two terms: i) the typical $O(\log t/t)$ guarantee known from the i.i.d. case, and ii) a term that depends on the amount of nonstationarity in the data input model. Our results recover the classical bounds under i.i.d. data input as a special case.

**Theoretical fairness and efficiency guarantees of PACE for nonstationary item arrivals.**

We consider the online fair allocation problem where item arrivals follow any of the three data input models that we consider for DA; these settings generalize the i.i.d. setting in Gao et al. (2021). Utilizing our convergence results for DA under nonstationary data input models, we show that, for item arrivals following these models, PACE ensures convergence of the pacing multipliers, again with a decomposition into a $O(\log t/t)$ term as well as a term depending on the nonstationarity. We then show that the agents' realized utilities, envy, regrets, and expenditures all obtain convergence bounds based on the convergence of pacing multipliers. Our results show that PACE as an online fair resource allocation algorithm is robust against distributional uncertainty of the input and automatically adapts to many different data input models without any parameter tuning. In Appendix E we provide numerical experiments which corroborate the above theory and demonstrate the practical efficiency of PACE under different data input models.

An extensive review of related work is provided in Appendix A.

**Notation.** We use $1_t$ to denote the vector of ones of length $t$ and $e_j$ to denote the vector with one in the $j$-th entry and zeros in the others. We use $\Delta(\Theta)$ to denote the space of probability measures on a measurable space $\Theta$, and $\Delta_n$ to denote the simplex in $\mathbb{R}^n$. To measure the nonstationarity in the input data, we will use the total variation distance. Given two probability measures $P$ and $Q$, it is defined as $\|P - Q\|_{\mathrm{TV}} := (1/2) \int |\frac{\mathrm{d}P}{\mathrm{d}\mu} - \frac{\mathrm{d}Q}{\mathrm{d}\mu}| \, \mathrm{d}\mu$, where $\mu$ is a supporting measure.

## 2 Preliminaries on Online Fair Allocation

An online fair allocation instance with infinitely divisible items with $n$ agents and a finite horizon $t$ consists of a tuple $\mathsf{A} = (n, t, \Theta, Q, v)$, where $\Theta$ is the (possibly uncountable) measurable space of all possible items, with an associated $\sigma$-algebra $\mathcal{M}$ and a probability measure $\mu$, the distribution $Q \in \Delta(\Theta^t)$ is the distribution over possible sequences of items $\gamma = (\theta_1, \ldots, \theta_t) \in \Theta^t$, each of unit supply, and the set $v = (v_1, \ldots, v_n) \in L^1_+(\Theta)^n$ is the set of valuation functions of the $n$ agents. Here $L^1_+(\Theta)$ is the space of positive integrable functions on $\Theta$. Agent $i$ sees a utility of $v_i(\theta)$ in item $\theta \in \Theta$. Abusing notation we let $v_i(\gamma) = \big( v_i(\theta^1), \ldots, v_i(\theta^t) \big)$ denote the valuation for agent $i$ of items in the sequence $\gamma$. Let $Q^\tau$ be the marginal distribution of the item $\theta^\tau$ at time $\tau$ and $\bar{Q} = (1/t)\sum_{\tau=1}^t Q^\tau$. We assume $\int_\Theta v_i \mathrm{d}\bar{Q} = 1$ for all $i \in [n]$. We further assume $|v|_\infty := \max_i \|v_i\|_\infty < \infty$. We stress that the PACE algorithm that we study is not going to require access to either the valuation functions $v$ or the set of possible items $\Theta$; these are only required in order to discuss the resulting bounds.

Given an instance $\mathsf{A}$, the decision maker allocates the stream of items $\gamma$ one at a time, in an irrevocable manner. At time $\tau$ when item $\theta_\tau$ is revealed, the decision maker must choose an allocation $x^\tau = (x_1^\tau, \ldots, x_n^\tau) \in \Delta_n$ based on information available at that time, and allocate accordingly. Here the $i$-th entry of $x^\tau$ is the fraction of item $\theta^\tau$ allocated to agent $i$. On receiving her fraction, agent $i$ realizes a utility of $u_i^\tau := v_i(\theta^\tau)x_i^\tau$. We let $x = (x^1, \ldots, x^t)$ denote the sequence of allocations made over time. For agent $i$, let $x_i = (x_i^1, \ldots, x_i^t) \in \mathbb{R}^t$ denote the fraction of items given to agent $i$ across time. With this notation, the total utility of agent $i$ is $\langle x_i, v_i(\gamma) \rangle$. The goal of the decision maker is to choose, in an online fashion, an allocation $x$ such that it achieves some form of both efficiency and fairness guarantees.

### 2.1 Benchmark: The Hindsight Allocation

As a benchmark, we will consider the hindsight-optimal allocation. Suppose all items are presented to the decision maker as opposed to arriving one by one. In that case, a fair and efficient allocation can be found by allocating using the *Eisenberg-Gale* (EG) convex program Eisenberg and Gale (1959). EG picks the allocation that maximizes the sum of weighted logarithmic utilities (which is equivalent to maximizing the weighted geometric mean of utilities):

$$\max_{x \geq 0, u \geq 0} \left\{ \sum_{i=1}^n B_i \log(U_i) \;\middle|\; U_i \leq \langle v_i(\gamma), x_i \rangle \;\; \forall i \in [n], \;\; \sum_{i=1}^n x_i^\tau \leq 1 \;\; \forall \tau \in [t] \right\}. \tag{1}$$

The weights $B_i$ represent the priority given to each agent, and they they can be interpreted as budgets in a market-based interpretation of the EG allocation.[1] We will focus on the case where $B_i = t/n$ for all $i$, but all our results extend directly to the case of unequal weights, which can be useful in settings such as when buyers have quasilinear utilities Gao et al. (2021); Conitzer et al. (2019) or when it is desirable to give a larger allocation to certain agents.

The PACE algorithm asymptotically converges to the optimal dual solution, which is

$$\beta^\gamma := \arg\min_{\beta \geq 0} \left\{ \frac{1}{t} \sum_{\tau=1}^t \max_{i \in [n]} \beta_i v_i(\theta^\tau) - \frac{1}{n} \sum_{i=1}^n \log \beta_i \right\}. \tag{2}$$

We will also be interested in the *underlying problem* implied by the average item supplies $s = \mathrm{d}\bar{Q}/\mathrm{d}\mu$. Letting $\langle v_i, x_i \rangle := \int_\Theta v_i x_i \, \mathrm{d}\mu$, this leads to the infinite-dimensional analogue of (1):

$$\max_{x \in L^\infty_+(\Theta), u \geq 0} \left\{ \frac{1}{n} \sum_{i=1}^n \log(u_i) \;\middle|\; u_i \leq \langle v_i, x_i \rangle \;\; \forall i \in [n], \;\; \sum_{i=1}^n x_i \leq s \right\}, \tag{3}$$

We let $u^*$ denote the optimal utilities in Eq. (3). The infinite-dimensional analogue of the dual (2) is the following. For any $\delta_0 > 0$, The infinite-dimensional analogue of (2) is the following. For any $\delta_0 > 0$,

$$\beta^* := \arg\min_{\frac{1}{n(1+\delta_0)} \leq \beta \leq 1+\delta_0} \left\{ \int_\Theta \Big( \max_{i \in [n]} \beta_i v_i(\theta) \Big) \bar{Q}(\mathrm{d}\theta) - \frac{1}{n} \sum_{i=1}^n \log \beta_i \right\}. \tag{4}$$

---

[1] The hindsight allocation Eq. (1) can be interpreted as a competitive equilibrium from equal incomes (CEEI) in the corresponding Fisher market; see Appendix B for more details on this interpretation.

A rigorous mathematical treatment of the infinite-dimensional program can be found in Gao and Kroer (2021) and (Gao et al., 2021, Section 2). Note the additional constraint in Eq. (4) on $\beta$ does not affect the optimal solution since $1/n \leq \beta_i^* \leq 1$; see Lemma 1 in Gao and Kroer (2021).

It is well-known that the hindsight allocation generated by the EG program enjoys the following efficiency and fairness properties:

1. Pareto optimality: we cannot strictly increase any agent's utility without decreasing some other agents' utility.

2. Envy-freeness: each agent prefers their own allocation to that of any other agent: $\langle v_i(\gamma), x_i^* \rangle \geq \langle v_i(\gamma), x_k^* \rangle$ for all $k \neq i$.

3. Proportionality: every agent achieves at least as much utility as under the uniform allocation, i.e. $\langle v_i(\gamma), x_i^* \rangle \geq \langle v_i(\gamma), (1/n)1_t \rangle$.

Therefore the hindsight EG allocation is the gold standard that we assume the decision maker would use if she had known the sequence of items $\gamma$ in advance. However, in the online setting the decision maker does not know this sequence, and must therefore instead attempt to approximate an equally good allocation in online fashion.

For an item sequence $\gamma$, we let $x^\gamma$ denote the optimal hindsight allocation, which is an optimal solution to Eq. (1), and we denote the resulting total and average utility as

$$U_i^\gamma := \langle v_i(\gamma), x_i^\gamma \rangle = \sum_{\tau=1}^{t} x_i^{\gamma,\tau} v_i(\theta^\tau), \quad u_i^\gamma := (1/t) \cdot U_i^\gamma . \tag{5}$$

## 2.2 Performance Metrics

We measure the performance of an online allocation rule $x$ on the instance $\gamma$ via the following two quantities. The *regret* of agent $i$ is the difference between the total hindsight equilibrium utility $U_i^\gamma$ and their realized utilities $u_i^\tau$ under $x$

$$\text{Reg}_{i,t}(\gamma) := U_i^\gamma - \sum_{\tau=1}^{t} u_i^\tau . \tag{6}$$

The *envy* of agent $i$ is the maximal extent to which they prefer the allocation of any other agent:

$$\text{Envy}_{i,t}(\gamma) := \max_{k \in [n]} \left\{ \langle v_i(\gamma), x_k \rangle - \langle v_i(\gamma), x_i \rangle \right\} . \tag{7}$$

We seek to understand the worst-case behavior of an algorithm when facing a certain class of input distributions. For a given input distribution class $\mathsf{C} \subset \Delta(\Theta^t)$, we will develop bounds on the worst-case regret and envy under any distribution in $\mathsf{C}$:

$$\sup_{Q \in \mathsf{C}} \mathbb{E}_{\gamma \sim Q} \left[ \text{Reg}_{i,t}(\gamma) \right] , \quad \sup_{Q \in \mathsf{C}} \mathbb{E}_{\gamma \sim Q} \left[ \text{Envy}_{i,t}(\gamma) \right] .$$

## 2.3 The PACE Algorithm

In this section, we review the PACE (Pace According to Current Estimated Utility) dynamics (Gao et al., 2021). In PACE, each item is allocated via first-price auction, and each agent constructs bids by scaling their value by a *pacing multiplier*. The pacing multipliers are maintained using simple, distributed updates that can be handled either by the agents or by the platform.

Algorithmic details are displayed in Algorithm 1. Here $\Pi_{[\ell,h]}[x] = \max\{\ell, \min\{x, h\}\}$. At every time step $\tau$ an item $\theta^\tau$ is revealed. At that point every agent comes up with a *bid* for that item, which is equal to their value for the item multiplied by their current pacing multiplier $\beta_i^\tau$. Then, the agents submit these bids to a first-price auction, and the item is allocated to the highest bidder. For concreteness, we choose the bidder with the smallest index if a tie occurs, but any rule works. Each agent then observes their realized utility, updates their average utility received so far, and updates their pacing multiplier accordingly. As pointed out in Gao et al. (2021), PACE is an instantiation of dual averaging Xiao (2010) applied to the dual of the hindsight allocation program in (4).

PACE has many attributes desirable in real-world applications.

---

**Algorithm 1:** PACE$(n, t, \delta_0)$

---

**Input:** number of agents $n$, horizon $t$, algorithm parameter $\delta_0 > 0$.

1 **Initialize:** Set $\beta^1 = (1 + \delta_0) \cdot 1_n$.

2 Environment draws the item sequence $\gamma = \{\theta^1, \ldots, \theta^t\}$ from the distribution $Q$.

3 **for** $\tau = 1, \ldots, t$ *when item $\theta^\tau$ is revealed* **do**

4     Agent $i$ bids $\beta_i^\tau v_i(\theta^\tau)$, the whole item $\theta^\tau$ is allocated to the highest bidder $i^\tau$ (with arbitrary tie breaking) $i^\tau := \min\{\arg\max_{i \in [n]} \beta_i^\tau v_i(\theta^\tau)\}$.

5     Agent $i$ updates current average utility $u_i^\tau = v_i(\theta^\tau) \mathbb{1}\{i = i^\tau\}$,    $\bar{u}_i^\tau = \frac{1}{\tau}\sum_{s=1}^\tau u_i^s$.

6     Agent $i$ updates the pacing multiplier $\beta_i^{\tau+1} = \Pi_{[\ell, h]}\left[1/(n\bar{u}_i^\tau)\right]$, where the interval
    $[\ell, h] = \left[\frac{1}{(1+\delta_0)n}, 1 + \delta_0\right]$.

---

**Highlight 1. Decentralization.** The PACE dynamics can be run in either centralized (by having the mechanism designer emulate the pacing process for each agent) or decentralized fashion (since the auction-based allocation is the only centralized step at each iteration), and are therefore suitable for Internet-scale online fair division and online Fisher market applications.

**Highlight 2. Pure Allocation.** PACE allows each item to be fully allocated to a single agent, even though the hindsight performance metric is allowed to utilize fractional allocations. While fractional allocations can be interpreted as randomized allocations in many large-scale settings, this may not always be desirable, for example when allocating food to food banks.

**Highlight 3. Tuning-free.** An important fact about the PACE dynamics is that each agent has no stepsize parameter whatsoever, which means that no stepsize tuning is required. Moreover, PACE is robust against *the types* of item arrivals since the algorithm needs neither knowledge of the item distribution $P$ nor the input type $\mathsf{C}$.

In addition to the regret and the envy performance metrics, we will also derive results for the following two quantities that characterize the long-run behavior of PACE. Let $\bar{u}^t = (1/t) \cdot \sum_{\tau=1}^t u^\tau$ be the vector of average realized utilities for all agents. We will show that the agents' utilities converge to those associated to the *underlying offline fair allocation problem*, $u^*$, defined in Eq. (3), in a mean-square sense, i.e., $\mathbb{E}\left[\|\bar{u}^t - u^*\|^2\right] \to 0$, as long as the error due to nonstationarity grows sublinearly in the number of time periods. Secondly, define the *expenditure* of agent $i$ at time $\tau$ by $b_i^\tau := \beta_i^\tau v_i(\theta^\tau) \mathbb{1}\{i = i^\tau\}$. We will show $(1/t) \cdot \sum_{\tau=1}^t b_i^\tau \to 1/n$ in mean square as well, as long as the error due to nonstationarity grows sublinearly in the number of time periods.

# 3 Main Results

This section introduces the main results of this paper: the behavior of PACE under three different types of nonstationary input models. All prior results on fair online allocation have been either for worst-case inputs (with much more conservative guarantees and not for the PACE algorithm) (Azar et al., 2016; Banerjee et al., 2022) or for i.i.d. input data Gao et al. (2021).

We first introduce some notation that will be useful for describing these input models. For $s > \tau \geq 1$ let $Q^s(\theta^{1:\tau})$ denote the conditional distribution of $\theta^s$ given $\{\theta^1, \ldots, \theta^\tau\}$. For a subset $I$ of $[t]$ let $Q^I$ denote the joint distribution of the variables $\{\theta^\tau\}_{\tau \in I}$. Let $\bar{Q} = (1/t) \cdot \sum_{\tau=1}^t Q^\tau$ be the uniform mixture of $\{Q^\tau\}_\tau$. We study three types of input: independent input with adversarial corruption, ergodic and Markov input, and periodic input. For each input setting, we describe our main theorem for the performance guarantees of PACE here. The proofs are given in Appendix C, because these results rely on developing a theory of nonstationary performance of DA, which is done in Section 4.

## 3.1 Independent Input with Adversarial Corruption

Adversarial perturbation of a fixed item distribution models scenarios where the items generally behave in a predictable manner, but for some time steps the input behaves erratically. Typically this is assumed to happen only for a small number of time steps. Such perturbation could be malicious,

for example when item arrivals are manipulated in favor of certain agents; or non-malicious, such as unpredictable surges of certain keywords on search engines (Estandiari et al., 2018).

We study a type of adversarial perturbation where the item distribution at each time step might be corrupted by an arbitrary amount, but distributions at different time steps are independent of each other. We assume the average corruption is bounded by $\delta$, as measured in TV distance. The set of distributions over sequences that we consider is then:

$$\mathsf{C}^{\mathrm{ID}}(\delta) := \left\{ Q \in \Delta(\Theta)^t : \frac{1}{t}\sum_{\tau=1}^t \|Q^\tau - \bar{Q}\|_{\mathrm{TV}} \le \delta \right\} . \tag{8}$$

We use $\tilde{O}$ to hide numeric constants and polynomials of $n$, $|v|_\infty$, and $\log t$. Our main fair online allocation result for the adversarial corruption case is:

**Theorem 1** (Independent Case). *For the adversarially corrupted and independent case, Algorithm 1 guarantees that for an instance* $\mathsf{A} = (n, t, \Theta, Q, v)$*, we have*

$$\sup_{Q \in \mathsf{C}^{\mathrm{ID}}(\delta)} \mathbb{E}_{\gamma \sim Q}\big[\mathrm{Reg}_{i,t}(\gamma)\big] \,, \ \sup_{Q \in \mathsf{C}^{\mathrm{ID}}(\delta)} \mathbb{E}_{\gamma \sim Q}\big[\mathrm{Envy}_{i,t}(\gamma)\big] = \tilde{O}\big(\sqrt{t} + \sqrt{\delta} \cdot t\big) \tag{9}$$

*and*

$$\sup_{Q \in \mathsf{C}^{\mathrm{ID}}(\delta)} \mathbb{E}_{\gamma \sim Q}\big[\|\bar{b}^t - (1/n)1_n\|^2\big] \,, \sup_{Q \in \mathsf{C}^{\mathrm{ID}}(\delta)} \mathbb{E}_{\gamma \sim Q}\big[\|\bar{u}^t - u^*\|^2\big] \,, \sup_{Q \in \mathsf{C}^{\mathrm{ID}}(\delta)} \mathbb{E}_{\gamma \sim Q}\big[\|\bar{u}^t - u^\gamma\|^2\big] = \tilde{O}(\delta + 1/t) \,.$$

The result shows that the regret and envy performance metrics degrade linearly in the average corruption $\delta$. In the i.i.d. case where $\delta = 0$, we recover the $\sqrt{t}$ regret rate and the $1/t$ rate of convergence for utilities and expenditures in terms of the mean-square error from Gao et al. (2021). If out of the $t$ distributions of items in each time step only $O(\sqrt{t})$ are corrupted, each by a constant amount, then the $\sqrt{t}$ regret and envy bounds, as well as $1/t$ convergence rates, are also preserved.

## 3.2 Ergodic Input and Markov Processes

To handle correlation across time, we next study ergodic inputs. For these inputs, strong correlation might be present for items sampled at nearby time steps, but the correlation between items decays as they are separated in time. For any integer $\iota$ such that $1 \le \iota \le t-1$, we measure the $\iota$-step deviation from some distribution $\Pi \in \Delta(\Theta)$ by the quantity

$$\delta(\iota) := \sup_\gamma \sup_{\tau=1,\dots,t-\iota} \|Q^{\tau+\iota}(\theta^{1:\tau}) - \Pi\|_{\mathrm{TV}} \,.$$

Intuitively, this definition tells us that, no matter where and when we start the item arrival process, it takes only $\iota$ steps to get $\delta(\iota)$-close to the distribution $\Pi$. We will consider the set of ergodic input distributions whose $\iota$-step deviation is bounded by $\delta$:

$$\mathsf{C}^{\mathrm{E}}(\delta, \iota) := \left\{ Q \in \Delta(\Theta^t) : \sup_\gamma \sup_{\tau=1,\dots,t-\iota} \|Q^{\tau+\iota}(\theta^{1:\tau}) - \Pi\|_{\mathrm{TV}} \le \delta, \text{ for some } \Pi \in \Delta(\Theta) \right\} . \tag{10}$$

**Theorem 2** (Ergodic Case). *For the ergodic case, Algorithm 1 guarantees that for an instance* $\mathsf{A} = (n, t, \Theta, Q, v)$*, we have*

$$\sup_{Q \in \mathsf{C}^{\mathrm{E}}(\delta, \iota)} \mathbb{E}_{\gamma \sim Q}\big[\mathrm{Reg}_{i,t}(\gamma)\big] \,, \ \sup_{Q \in \mathsf{C}^{\mathrm{E}}(\delta, \iota)} \mathbb{E}_{\gamma \sim Q}\big[\mathrm{Envy}_{i,t}(\gamma)\big] = \tilde{O}\big(\sqrt{\iota t} + \sqrt{\delta} \cdot t\big) \tag{11}$$

*and*

$$\sup_{Q \in \mathsf{C}^{\mathrm{E}}(\delta, \iota)} \mathbb{E}_{\gamma \sim Q}\big[\|\bar{b}^t - (1/n)1_n\|^2\big] \,, \sup_{Q \in \mathsf{C}^{\mathrm{E}}(\delta, \iota)} \mathbb{E}_{\gamma \sim Q}\big[\|\bar{u}^t - u^*\|^2\big] \,, \sup_{Q \in \mathsf{C}^{\mathrm{E}}(\delta, \iota)} \mathbb{E}_{\gamma \sim Q}\big[\|\bar{u}^t - u^\gamma\|^2\big] = \tilde{O}(\delta + \iota/t) \,.$$

**Remark 1** (Markov Input). *We can specialize the result in Theorem 2 to fast mixing or Markov item sequences. Fast mixing means the deviation $\delta$ decreases exponentially, i.e., for all $1 \le \iota \le t-1$,*

$$\sup_\gamma \sup_{\tau=1,\dots,t-\iota} \|Q^{\tau+\iota}(\theta^{1:\tau}) - \Pi\|_{\mathrm{TV}} \le M\rho^\iota \,, \tag{12}$$

*for some $M > 0$, $\rho \in [0, 1)$, and $\Pi$ is the stationary distribution. Examples include finite state-space time-homogeneous Markov chains and uniformly ergodic Markov chains on general state spaces (Meyn and Tweedie, 2012, Chapter 16). In these cases, setting $\iota = \frac{\log t + \log M}{\log(\rho^{-1})} = O\big(\frac{\log t}{\log(\rho^{-1})}\big)$ implies $\delta \le 1/t$. This means the Markov chain from which $\gamma$ is generated takes $O(\log t)$ steps to get $(1/t)$-close to stationarity. The dominant term for the regret in Theorem 2 (further ignoring $M$) is then $(1 + \frac{1}{\log(\rho^{-1})})^{1/2}\sqrt{t}$. The term in the parenthesis reflects the inflation caused by input dependency. To recover the case of i.i.d. input, we simply send $\rho \to 0$ and the usual $\sqrt{t}$ regret and envy rates and $1/t$ utility and expenditure convergence rates are again recovered.*

## 3.3 Periodic Input

Item sequences often exhibit statistical periodic structure. For example, when allocating compute time to requestors, there will be more requests during weekdays and less on weekends. The compute request patterns vary throughout the week, and yet the weekly pattern would repeat over time. Similarly, Internet traffic typically exhibits periodic structure.

Formally, assume that the horizon $t$ divides into $K$ blocks of time, each of size $q$. This divides the item sequence $\gamma$ into consecutive blocks of length $q$. Within each block, we allow a distribution with arbitrary dependence between time steps, but we assume that the blocks, as a whole, are identically and independently distributed. We define the set of periodic input distributions as follows:

$$\mathsf{C}^{\mathrm{P}}(q) := \left\{ Q \in \Delta(\Theta^q)^K : Q^{1:q} = Q^{q+1:2q} = \ldots = Q^{t-q+1:t} \right\}. \tag{13}$$

**Theorem 3** (Periodic Case)**.** *For the periodic case, Algorithm 1 guarantees that for an instance* $\mathsf{A} = (n, t, \Theta, Q, v)$, *we have*

$$\sup_{Q \in \mathsf{C}^{\mathrm{P}}(q)} \mathbb{E}_{\gamma \sim Q} \left[ \mathrm{Reg}_{i,t}(\gamma) \right], \ \sup_{Q \in \mathsf{C}^{\mathrm{P}}(q)} \mathbb{E}_{\gamma \sim Q} \left[ \mathrm{Envy}_{i,t}(\gamma) \right] = \tilde{O}\left( \sqrt{qt} \right) \tag{14}$$

*and*

$$\sup_{Q \in \mathsf{C}^{\mathrm{P}}(q)} \mathbb{E}_{\gamma \sim Q} \left[ \|\bar{b}^t - (1/n)1_n\|^2 \right], \sup_{Q \in \mathsf{C}^{\mathrm{P}}(q)} \mathbb{E}_{\gamma \sim Q} \left[ \|\bar{u}^t - u^*\|^2 \right], \sup_{Q \in \mathsf{C}^{\mathrm{P}}(q)} \mathbb{E}_{\gamma \sim Q} \left[ \|\bar{u}^t - u^\gamma\|^2 \right] = \tilde{O}(q^2/t).$$

If the length $q$ of the blocks is of order $o(t)$ then the time-averaged regret and envy are both vanishing. For the i.i.d. case, we can set $q = 1$ to recover the previous results.

Now we comment on dependence on the period length $q$. Suppose the item sequence consists of $K$ blocks, and blocks are i.i.d. We still allow arbitrary dependence within a block. The proof of Theorem 3 essentially relies on the result (Theorem 9) that DA produces iterates whose squared error is of order $O(q^2/t)$. Consider dual averaging with the knowledge of the block structure $q$. Then the rate $1/K = q/t$ can be achieved by executing DA using one randomly chosen data point within a block, throwing away the rest in that same block. Such selection produces $K$ i.i.d. samples from the distribution. In comparison, the rate in $O(q^2/t)$ is worse off by a factor of $q$ due to not knowing the block-structure information.

## 4 Proof Technique: Nonstationary Dual Averaging

The PACE dynamics can be cast as online dual averaging (Xiao, 2010) applied to the dual of the hindsight allocation program in (1). This will be discussed in more detail in Section 4.2 and Appendix C.2. However, in order to characterize its performance under various types of nonstationary input, we first extend the general convergence results for dual averaging to incorporate nonstationary input. The convergence results that we developed for dual averaging under three different types of input models are novel and are of independent interest.

We remark that the results for the stochastic setting given by Xiao (2010) cannot be used directly, since they rely on the stringent i.i.d. assumption. Duchi et al. (2012) consider ergodic mirror descent (MD) for convex problems under some (but not all) of our nonstationary input models. Their results and analysis cannot be used in our case either, since their results do not allow using the composite structure, whose strong convexity we leverage. Moreover, unlike DA, an MD-based approach would require tuning parameters such as stepsizes.

In this section, after introducing the setup of DA in Section 4.1, we present a DA convergence result for independent but not identical input in Section 4.3, for which we outline the proof idea and technical challenges in Section 4.3. Due to space limitations, we present DA convergence results for ergodic and periodic inputs in Appendix D.3.

In the nonstationary setup of DA, we emphasize that whenever we mention convergence, we mean convergence of DA iterates to the population-level optimum $w_\Pi^*$ (or sometimes the hindsight optimum), up to some error caused by nonstationarity.

## 4.1 Review: The DA Algorithm

We review the dual averaging setup in the strongly convex case (Xiao, 2010, §1.1). Consider a stochastic optimization problem of the form

$$\min_w \left\{ \phi(w) := \mathbb{E}_{z \sim \Pi}\big[F(w, z)\big] = \mathbb{E}_{z \sim \Pi}\big[f(w, z)\big] + \Psi(w) \right\}, \tag{15}$$

where $w \in (\mathbb{R}^d, \|\cdot\|)$ is the variable, $\Psi$ is a closed convex function with closed domain $\mathrm{Dom}\,\Psi := \{w \in \mathbb{R}^n : \Psi(w) < \infty\}$. The expectation is taken over a probability distribution $\Pi$ on a measurable space $Z$. For each $z \in Z$, the function $f(\cdot, z)$ is convex and subdifferentiable (a subgradient always exists) on $\mathrm{Dom}\,\Psi$. Let $F(w, z) = f(z, w) + \Psi(w)$.

**Assumptions.** Let $\mathsf{G}(w, z)$ be a fixed element in the set of subgradients $\partial_w f(w, z)$.

1. For almost every $z$, it holds $\|\mathsf{G}(w, z)\|_* \leq G$, where $\|\cdot\|_* = \max_{\|w\| \leq 1} \langle s, w \rangle$ is the dual norm.

2. There exists an $\bar{F} \in \mathbb{R}$ such that $F(w, z) \leq \bar{F}$ for all $w \in \mathrm{Dom}\,\Psi$ and (almost every) $z$.

3. $\Psi$ is $\sigma$-strongly convex, i.e., $\Psi(\alpha w + (1 - \alpha)u) \leq \alpha\Psi(w) + (1 - \alpha)\Psi(u) - \frac{\sigma}{2}\alpha(1 - \alpha)\|w - u\|^2$ for $w, u \in \mathrm{Dom}\,\Psi$.

Because of our strong convexity assumption, the solution to (15) is unique. Associated with $\Pi$ we define $w_\Pi^* := \arg\min \mathbb{E}_{z \sim \Pi}\big[F(w, z)\big]$.

The dual averaging algorithm (DA) (Xiao, 2010, Algorithm 1) aims to produce a sequence converging to the optimal point $w_\Pi^*$ or minimize the associated regret (Xiao, 2010, §1.2). The algorithmic details for DA are presented in Algorithm 2. Note that although Xiao (2010) only considers the case of i.i.d. data, DA iterates are defined for every input sequence $\{z_\tau\}_{\tau=1}^t$, regardless of any distributional properties of the sequence.

The first step in our analysis is a relationship between regret and the suboptimality $\|w_t - w_\Pi^*\|$ derived by Xiao (2010). Consider the dual averaging algorithm with data $\{z_\tau\}_{\tau=1}^t$. We denote the one-step subgradient by $g_\tau := \mathsf{G}(w_\tau, z_\tau)$ and the average subgradient by $\bar{g}_\tau = (\sum_{s=1}^\tau g_s)/\tau$. Given data $\{z_\tau\}_{\tau=1}^t$, we define the regret and the sum of squared subgradient norms [2]

$$R_t(w) := \sum_{\tau=1}^t \big(F(w_\tau, z_\tau) - F(w, z_\tau)\big), \quad \Delta_t := \frac{1}{2\sigma}\big(5\|g_1\|_*^2 + \sum_{\tau=1}^{t-1} \|g_{\tau+1}\|_*^2/\tau\big).$$

The bound $\Delta_t \leq (6 + \log t)G^2/(2\sigma)$ holds in a deterministic manner due to the bounded subgradient assumption.

Xiao (2010) shows the following bound on suboptimality of $w_t$

**Lemma 1** (Regret Bound, Section B.2 in Xiao (2010)). *For any sequence $\{z_\tau\}_{\tau=1}^t$, any $w \in \mathrm{Dom}\,\Psi$, any $t = 1, 2, \ldots$, it holds $\|w_{t+1} - w\|^2 \leq \frac{2}{\sigma t}\big(\Delta_t - R_t(w)\big)$.*

## 4.2 Review: PACE as Dual Averaging

In this section we review how to cast PACE as dual averaging applied to the problem (4). This derivation was originally given in Gao et al. (2021). Let $f(\beta, \theta) = \max_i \beta_i v_i(\theta)$, $\Psi(\beta) = -\frac{1}{n}\sum_{i=1}^n \log \beta_i$, in which case we get $F(\beta, \theta) = f(\beta, \theta) + \Psi(\beta) = \max_{i \in [n]} \{\beta_i v_i(\theta)\} - \frac{1}{n}\sum_{i=1}^n \log \beta_i$. Following (Gao and Kroer, 2021, §5), since $f(\cdot, \theta)$ is a piecewise linear function, a subgradient is $\mathsf{G}(\beta, \theta) := v_{i^\tau}(\theta)e_{i^\tau} \in \partial_\beta f(\beta, \theta)$, where $i^\tau = \min\{\arg\max_i \beta_i v_i(\theta)\}$ is the index of the winning agent (see, e.g., Beck (2017, Theorem 3.50)). Based on this instantiation of DA, we get that the iterates $\{\beta^\tau\}_{\tau=1}^{t+1}$ generated by the PACE dynamics (Algorithm 1) are exactly the iterates $w_\tau$ generated by $\mathrm{DA}(\mathsf{G}, \Psi, \gamma)$ (Algorithm 2). A proof is given in Appendix C.2.

Before moving on to showing our results, let us first touch on the fact that dual averaging provides worst-case regret guarantees. Naively, one may expect that these regret guarantees would directly translate into regret guarantees on the primal performance, meaning a bound on $\sup_\gamma \mathrm{Reg}_{i,t}(\gamma)$.

---

[2] See the first equation on page 2584 in Xiao (2010). In Xiao (2010)'s notation, set $\beta_\tau = 0$ all $\tau \geq 1$ and $\beta_0 = \sigma$, plug in the bound $h(w_2) \leq 2\|g_1\|_*^2/\sigma$ and we have the expression of $\Delta_t$ in our paper.

**Algorithm 2:** $\mathsf{DA}(\mathsf{G}, \Psi, \{z_\tau\}_{\tau=1}^t)$

---

**Input:** subgradient $\mathsf{G}$, regularizer $\Psi$ and data $\{z_\tau\}_{\tau=1}^t$ .
1 **Initialize:** set $\bar{g}_0 = 0$ and $w_1 = \arg\min \Psi$.
2 **for** $\tau = 1, \ldots, t$ **do**
3      Observe $z_\tau$ and compute $g_\tau = \mathsf{G}(w_\tau, z_\tau)$.
4      Average subgradients (the *dual average*) via $\bar{g}_\tau = \frac{\tau-1}{\tau}\bar{g}_{\tau-1} + \frac{1}{\tau}g_\tau$.
5      Compute the next iterate $w_{\tau+1} = \arg\min_w\{\langle \bar{g}_\tau, w\rangle + \Psi(w)\}$.

---

However, such dual regret bounds do not imply a worst-case primal regret bound. From a technical perspective, it is unclear how a regret bound on the dual objective would translate to a regret bound on envy or utilities. Secondly, based on results in the online fair allocation literature (Azar et al., 2016; Banerjee et al., 2022), it is known that not only can we not get a no-regret guarantee on the primal performance, it is not even possible to achieve a constant competitive ratio in the worst-case setting.

### 4.3 Independent Data with Adversarial Corruption

We present a DA convergence result with independent data in this section. Theorem statements for the ergodic case and periodic case are presented in Appendix D.3. Discarding the i.i.d. assumption on the data $\{z_\tau\}_{\tau=1}^t$, we let $P$ be the joint distribution of $\{z_\tau\}_\tau$ and let $P^\tau$ be the marginal distribution of $z_\tau$. We study the relationship between the DA iterates $w_{t+1}$ and $w_\Pi^*$ by bounding the mean-square difference $\mathbb{E}_{\{z_\tau\}_{\tau=1}^t \sim P}\big[\|w_{t+1} - w_\Pi^*\|^2\big]$, and thus demonstrate in what sense the data distribution $P$ should stay close to the i.i.d. distribution $\Pi$ in order to preserve DA convergence.

We first introduce a variant of $\mathsf{C}^{\mathrm{ID}}(\delta)$ with a target distribution $\Pi$:

$$\mathsf{C}^{\mathrm{ID}}(\delta; \Pi) := \left\{ P \in \Delta(\Theta)^t : \frac{1}{t}\sum_{\tau=1}^t \|P^\tau - \Pi\|_{\mathrm{TV}} \le \delta \right\}. \tag{16}$$

**Theorem 4** (DA Convergence, Independent Data with Adversarial Corruption). *If $\{z_\tau\}_{\tau=1}^t \sim P$ and $P \in \mathsf{C}^{\mathrm{ID}}(\delta, \Pi)$. Then for $t \ge 1$,*

$$\mathbb{E}_{\{z_\tau\}_{\tau=1}^t \sim P}\big[\|w_{t+1} - w_\Pi^*\|^2\big] \le \frac{(6 + \log t)G^2}{\sigma^2 t} + \frac{8\bar{F}}{\sigma}\delta = \tilde{O}(\delta + 1/t) .$$

*Moreover, the rate $\tilde{O}(\delta + 1/t)$ applies to $\mathbb{E}\big[\|w_{t+1} - w_\gamma^*\|_2^2\big]$ and $\mathbb{E}\big[\|w_\gamma^* - w_\Pi^*\|_2^2\big]$ (See Appendix D.1).*

**Remark 2.** *From this result we can tell when DA retains last-iterate convergence. Suppose the number of corrupted data points is of order $o(t)$ (assuming corruption on each data is of the same order), then the corruption per item is $\delta = o(1)$ and DA converges to the optimal solution as $t \to \infty$. Furthermore, if the corruption per data point is of order $\delta = O(1/t)$, then the fast rate $1/t$ is retained.*

By Section 4.2, we get as a corollary that the PACE iterates $\{\beta_t\}$ will converge to $\beta^*$, the solution to the infinite-dimensional dual program (4). This is the building block for our results in Section 3.

While the full proof is too long to fit in the paper, we now give a proof sketch to show the main ideas behind how we derive Theorem 4. We begin the proof by noticing Lemma 1 is deterministic and valid for any $\{z_\tau\}_{\tau=1}^t$. Now set $w = w_\Pi^*$ in Lemma 1. If the input data $\{z_\tau\}_{\tau=1}^t$ were i.i.d. from $\Pi$, i.e., $P = \Pi^{\otimes t}$, then $\mathbb{E}[R_t(w_\Pi^*)]$ would be greater than zero, and we would obtain

$$\mathbb{E}\big[\|w_{t+1} - w_\Pi^*\|^2\big] \le \frac{2}{\sigma t}\Big(\mathbb{E}[\Delta_t] - \mathbb{E}\big[R_t(w_\Pi^*)\big]\Big) \le \frac{(6 + \log t)G^2}{\sigma^2 t} .$$

However, in the nonstationary case, the regret $\mathbb{E}[R_t(w_\Pi^*)]$ might be negative. At a high level, our results are achieved by introducing appropriate measures of the nonstationarity and then lower bounding $\mathbb{E}[R_t(w_\Pi^*)]$ based on those measures.

To this end, we decompose the regret as follows. Write

$$R_t(w_\Pi^*) = \underbrace{\sum_{\tau=1}^t \left(F(w_\tau, z_\tau) - \phi_\Pi(w_\tau)\right) + \sum_{\tau=1}^t \left(\phi_\Pi(w_\Pi^*) - F(w_\Pi^*, z_\tau)\right)}_{\text{I}} + \underbrace{\sum_{\tau=1}^t \left(\phi_\Pi(w_\tau) - \phi_\Pi(w_\Pi^*)\right)}_{\text{II}} .$$

By optimality of $w_\Pi^*$ we have $\text{II} \geq 0$. Using the bound on the TV distance between $\{P^\tau\}_\tau$ and $\Pi$, and boundedness of $F$ we can control the other two terms. The key is, conditional on $\mathcal{F}_{\tau-1}$, the iterate $w_\tau$ is deterministic and the distribution of $z_\tau$ is $P^\tau$ due to independence assumption. For each term in the first summation, we condition on $\mathcal{F}_{\tau-1}$ and obtain

$$
\begin{aligned}
|\mathbb{E}[F(w_\tau, z_\tau) - \phi_\Pi(w_\tau)|\mathcal{F}_{\tau-1}]| &= \left|\mathbb{E}\left[\int_{\mathcal{Z}} F(w_\tau, z)P^\tau(\mathrm{d}z \mid z_{1:\tau-1}) - \int_{\mathcal{Z}} F(w_\tau, z)\,\mathrm{d}\Pi(z)|\mathcal{F}_{\tau-1}\right]\right| \\
&= \left|\mathbb{E}\left[\int_{\mathcal{Z}} F(w_\tau, z)P^\tau(\mathrm{d}z) - \int_{\mathcal{Z}} F(w_\tau, z)\,\mathrm{d}\Pi(z)|\mathcal{F}_{\tau-1}\right]\right| \\
&\leq \mathbb{E}\left[\left|\int_{\mathcal{Z}} F(w_\tau, z)P^\tau(\mathrm{d}z) - \int_{\mathcal{Z}} F(w_\tau, z)\,\mathrm{d}\Pi(z)\right||\mathcal{F}_{\tau-1}\right] \\
&\leq 2\bar{F}\|P^\tau - \Pi\|_{\text{TV}} .
\end{aligned}
$$

The second sum can be handled similarly. For the detailed proof and a generalization see Appendix D.2 in Appendix D.

We now discuss how this paper handles nonstationarity differently from the existing literature. Let us use Balseiro et al. (2020) as a reference point, since they consider very similar categories of nonstationary inputs. From a online constrained optimization perspective, Balseiro et al. (2020) relies on the fact that their objective is separable across timesteps in order to handle nonstationarity. In contrast, this structure does not exist for the online fair allocation problem, because the objective takes the logarithm of the utility over time. Concretely, in Balseiro et al. (2020), the objective is of the form $\sum_{\tau=1}^t f_\tau(x)$. This type of time-separability occurs in all works that we are aware of for non-stationary inputs, also e.g. the ergodic mirror descent paper Duchi et al. (2012) (Duchi et al. (2012) is not an online setting, but the expectation in their objective is analogous to time separability). Time separability is used as a key property for deriving regret bounds in those papers. The separability enables translating dual regret to primal regret by a weak duality argument (see e.g. Prop. 1 in Balseiro et al. (2020) where a time-separable dual allows weak duality).

In contrast, our problem has time-separability only in the dual formulation, but not in the primal one, which is where we ultimately want guarantees (since we are interested in utilities converging). Our contribution to handling nonstationarity in online fair allocation is showing that a dual approach works even without the separability structure. We begin by deriving convergence guarantees on the last dual iterate, which we achieve by modifying the dual averaging proof to take into account nonstationarity and analyzing the stability of the dual variables in dual averaging. We note that this technique depends on strong convexity, unlike for the time-separable case. Next, to go from dual last-iterate convergence to primal regret bounds we use the first-order optimality condition for EG program, which are specific techniques for our problem (such techniques were also used in the previous PACE paper Gao et al. (2021)).

## 5 Conclusion

We established new convergence results for dual averaging under nonstationary data input models, namely, adversarial corruption, ergodic, and block-independent input models. Leveraging these results, we showed that, for online fair allocation problems with item arrivals generated from the above nonstationary data input models, the PACE algorithm automatically adapts to them and achieves asymptotic fairness and efficiency without any parameter tuning. Numerical experiments demonstrated the effectiveness of PACE under these data input models.

## Acknowledgments and Disclosure of Funding

This research was supported by the Office of Naval Research Young Investigator Program under grant N00014-22-1-2530, and the National Science Foundation award IIS-2147361.

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
