# OpenReview forum: "Nonstationary Dual Averaging and Online Fair Allocation"
_NeurIPS.cc/2022/Conference — NeurIPS 2022 Accept_

### Official Review · Reviewer_4sGJ · 2022-06-24

**Rating:** 6
**Confidence:** 1
**Soundness:** 4 excellent
**Presentation:** 4 excellent
**Contribution:** 4 excellent

**Summary:**

As I have notified the AC, this paper is certainly beyond my expertise. In addition, I’m not familiar with most paper in references. I'm simply making an educated guess.

**Questions:**

Is there any connection between the nonstationary setting in this paper and nonstationary reinforcement learning?

Ref:

Non-stationary reinforcement learning without prior knowledge: An optimal black-box approach

Nonstationary reinforcement learning with linear function approximation


**Strengths And Weaknesses:**

From an outsider view, I think this paper is clearly written.

The box around the reference is a little annoying, probably it is better to drop it.

---

> ### Author Response · Authors · 2022-08-03
> **response**
>
> Thank you for bringing these related works to our attention.
>
> We agree that there is some similarity in how we formalize the notion of nonstationarity compared to the papers you pointed out. However, the problem we deal with is very different from nonstationary RL.
>
> The difference is most significant when we look at how nonstationarity enters regret.
>
> In nonstationary episodic RL, nonstationarity enters in an _additive_ way. The benchmark is the sequence of optimal policies corresponding to the reward functions and transition probability functions over the episodes. For each episode, we find the difference in values between the employed policy and the optimal policy (for that episode), and then sum up such differences. While in our case the benchmark is essentially a convex program that depends on all observed data across time steps, so nonstationarity enters the benchmark in a coupled way. To handle this, we take a dual approach and develop a nonstationary theory for dual averaging.
>
> We thank you for pointing out these references and we will add them and the discussion above to the literature review section.

---

> > ### Comment · Reviewer_4sGJ · 2022-08-09
> > **thanks**
> >
> > Thanks for your response! That addressed my questions.

---

### Official Review · Reviewer_2nRt · 2022-07-07

**Rating:** 6
**Confidence:** 4
**Soundness:** 3 good
**Presentation:** 3 good
**Contribution:** 2 fair

**Summary:**

This paper extends PACE [Gao et al 2021]  for online fair allocation to nonstationary settings. In particular, adversarially-corrupted stochastic, ergodic and periodic input are studied. Technically, this is done by using the connection of PACE with dual averaging (DA), and the new derivations of DA regret.

**Questions:**

Typo: line 112 "they they"

**Limitations:**

The main application domain is related to fairness.

**Strengths And Weaknesses:**

(+) This work extends PACE, which was designed for stochastic problems, to non-stationary settings.

(-) The online learning version of DA, aka follow the regularized leader (FTRL, see section 5 of https://arxiv.org/pdf/1909.05207.pdf) is capable of handling adversarial settings. Hence, it is not surprising that DA or PACE works for the nonstationary setting studied in this work. Can the authors discuss more on this?

---

> ### Author Response · Authors · 2022-07-29
> **DA worst-case regret DOES NOT imply social-welfare/envy guarantees for fair allocation**
>
> > The online learning version of DA, aka follow the regularized leader (FTRL, see section 5 of https://arxiv.org/pdf/1909.05207.pdf) is capable of handling adversarial settings. Hence, it is not surprising that DA or PACE works for the nonstationary setting studied in this work. Can the authors discuss more on this?
>
> Response:
>
> First, whether a result is “surprising” or not is usually not considered a good way to measure whether a result is important. But in any case, what the reviewer is implying is perhaps something more along the lines of “this result seems more or less directly implied by adversarial results for DA/FTRL.” With all due respect, we think you misunderstood the relationship between regret bounds *on the dual* and the types of *primal* regret guarantees we are achieving in this paper.
>
> From a technical point of view, to obtain a guarantee on the performance metrics we desire, e.g., social welfare and envy (Section 2.2), we need more than just regret analysis of dual averaging; we need the last iterate convergence guarantee provided via strong convexity for the stochastic setting. As we explained briefly in Section 4.3 Line 321-326, a worst-case regret guarantee of DA does not necessarily imply a last iterate convergence in nonstationary environments (for example, negative regret could occur due to nonstationarity). In fact, one needs to carefully modify the dual averaging proof to take into account nonstationarity and analyze the stability of the dual variables in dual averaging (i.e., ensuring that the iterates do not change too much from one step to the next). As a side product, we address an open problem of investigating DA under Markov noise, proposed back in 2010 in [Section 7.3, Xiao 2010].
>
> Our results show that DA enjoys last-iterate convergence in the face of various types of nonstationarity. For example, in Theorem 1, our result shows that if only $o(t)$ number of items are corrupted, then the average corruption per item, $\delta = o(1)$ and we achieve no-regret again. Our results describe the robustness of DA against nonstationarity in a fine-grained manner, none of which are available in, or implied by, the reference you pointed out.
>
> Finally, as a more indirect argument to see why worst case regret guarantees on the dual are not directly meaningful, we emphasize that *primal* worst-case no-regret guarantees on buyer utilities are not even possible for online fair allocation with adversarial inputs. See the counterexample from (Section B of [a]). This is in spite of the adversarial guarantees achieved by DA.
>
> [a] Gao, Y., Kroer, C., and Peysakhovich, A. (2021). Online market equilibrium with application to fair division. NeurIPS 2021, https://proceedings.neurips.cc/paper/2021/hash/e562cd9c0768d5464b64cf61da7fc6bb-Abstract.html

---

> > ### Comment · Reviewer_2nRt · 2022-08-05
> > **Thanks for your response and some follow-up questions**
> >
> > Suppose that there are two submissions. One has important and interesting results, and the other is only important. I think the author will agree with me that the former has better chance to get accepted. I am not saying that your result is not important, yet I appreciate your theorems. However, the results here are only solid but not inspiring enough to me.
> >
> > Your response addressed part of my concerns, but I am still not perfectly convinced. Please find my questions below.
> >
> > Q1. The non-stationary considered in this paper is more or less disguised under the cover of stochasticity. Are there any guarantees on the last-iteration convergence of DA in (purely) stochastic setting?
> >
> > Q2. Although adversarially-corrupted stochastic, ergodic and periodic input are not considered in online fair allocation, most of them have been carefully investigated in other settings. Can the authors clarify more on your unique contributions on the methods for studying these non-stationary settings?

---

> > > ### Author Response · Authors · 2022-08-05
> > > **Thank you for your follow-up question**
> > >
> > > > Suppose that there are two submissions. One has important and interesting results, and the other is only important. I think the author will agree with me that the former has better chance to get accepted
> > >
> > > Sure; our concern was primarily that you labeled the result "not surprising" which is usually viewed as an unfortunate way to gauge paper quality. Whether a result is interesting or important are more useful labels to use, in our view.
> > >
> > > Q1: Regarding last-iterate convergence of DA in the purely stochastic setting: yes, such results are known. In fact, this is exactly the setting that the previous PACE paper worked in, and those results leveraged the DA last-iterate guarantees for the stochastic setting. We don’t agree, though, that the present results are “disguised under the cover of stochasticity.” The point of the paper (as well as e.g. [a, Section 5] and [b]), is that we would like to be able to guarantee good performance when we have an input process that lies somewhere between adversarial and stationary stochastic inputs. To do that, we essentially must have something that measures the degree to which the process is allowed to depart from stationary, since we know that if we allow unbounded deviation from stationary, then it can simulate adversarial inputs, where no guarantees are possible in our setting (without further assumptions as in the papers described  in the introduction).
> > >
> > > Q2: We agree they have been studied in other settings.
> > >
> > > Let us use [a] as a reference point, since they consider very similar categories of nonstationary inputs. From an online constrained optimization perspective, [a] relies on having a separable objective across timesteps to handle nonstationarity, while such a nice structure does not exist in the online fair allocation problem/Eisenberg-Gale program.
> > >
> > > Concretely, in [a], the objective is of the form $\sum_t f_t(x_t)$. This type of time-separability occurs in all works that we are aware of for non-stationary inputs, also e.g. the ergodic mirror descent paper [b] ([b] is not an online setting, but their expectation is analogous to time separability). It is used as a key part of how to derive regret bounds in those papers. The separability enables translating dual regret to primal regret simply by a weak duality argument (see e.g. Prop. 1 in [a] where a time-separable dual allows weak duality).
> > >
> > > In contrast, our problem has time-separability only in the dual space, but not in the primal space, which is where we ultimately want guarantees (since we are interested in utilities converging). This then goes back to our earlier comment to you: regret guarantees in the dual space do not lead to guarantees in the primal space. This is the core reason why we have to work with guarantees on the last iterate, as opposed to just regret as in other papers on non-stationarity.
> > >
> > > Our contribution to handling nonstationarity in online fair allocation is showing that a dual approach works even without the separability structure. To go from dual regret to primal guarantees we use the first-order optimality condition for EG program, which are specific techniques for our problem (such techniques were also used in the previous PACE paper [c]). To this end we need last-iterate convergence, which we achieved by modifying the dual averaging proof to take into account nonstationarity and analyzing the stability of the dual variables in dual averaging.
> > >
> > > We thank the reviewer for bringing Q2 to our attention: we should have discussed this distinction in the paper. We will definitely include this discussion in the revised version.
> > >
> > >
> > > [a] Balseiro, S., Lu, H., and Mirrokni, V. (2020). The best of many worlds: Dual mirror descent for online allocation problems. arXiv preprint arXiv:2011.10124.
> > >
> > > [b] Duchi, J. C., Agarwal, A., Johansson, M., and Jordan, M. I. (2012). Ergodic mirror descent. SIAM Journal on Optimization, 22(4):1549–1578.
> > >
> > > [c] Gao, Y., Kroer, C., and Peysakhovich, A. (2021). Online market equilibrium with application to fair division. NeurIPS 2021,

---

> > > > ### Comment · Reviewer_2nRt · 2022-08-06
> > > > **Concern addressed**
> > > >
> > > > Thanks for replying to my questions. After some clarifications on Q2, the contribution of this paper is more concrete to me. I am happy to increase the score accordingly.

---

### Official Review · Reviewer_nihF · 2022-07-09

**Rating:** 6
**Confidence:** 2
**Soundness:** 3 good
**Presentation:** 3 good
**Contribution:** 3 good

**Summary:**

This paper considered the problem of fair allocation of sequential items arrivals.  In particular, this paper extended the existing PACE algorithms, which requires i.i.d. item arrivals, to non-stationary cases.  To this end, this paper studied three non-stationary input models: adversarially-corrupted stochastic input, ergodic input, and block-independent input.  For each case, this paper presented new convergence results for PACE with dual averaging algorithm. Finally, numerical results are presented to validate the performance of the PACE against non-stationary inputs.

**Questions:**

See limitations above.

**Ethics Review Area:**

["I don’t know"]

**Limitations:**

N.A.

**Strengths And Weaknesses:**

Strength:
- Generalization of the PACE to practical setting, i.e., non-stationary data.  This is critical and makes PACE more applicable in practice.  In particular, three non-stationary models are investigated.
- Strong theoretical performance analysis:  For each of the non-stationary model, new convergence results are provided.  The convergence analysis for dual averaging under these three input models are novel and of independent interests.  This will benefit the community at large.

Limitations:

The reviewer does not identify any major issues in this paper but has some minor comments that can be found below.

Overall, this paper is well written and relatively easy to follow, despite its very theoretical nature. The proposed new ideas and algorithms are clearly presented.

- However, some good intuitions are missing in some parts for readers to better interpret the theoretical results, e.g., no discussions for Theorem 3 and Theorem 4. The paper seems to end unexpectedly after Theorem 4.
- The background of the paper goes almost to page 5, which does not leave enough space to present and discuss the main results in the paper.  It may be better to make the background concise and some may be moved to the supplementary material.
- The abstract mentioned the numerical results of the paper; however, the experiments are not presented, until the end of the supplementary material.

---

> ### Author Response · Authors · 2022-07-29
> **response**
>
> > However, some good intuitions are missing in some parts for readers to better interpret the theoretical results, e.g., no discussions for Theorem 3 and Theorem 4. The paper seems to end unexpectedly after Theorem 4.
>
> Response:
>
> We will add more discussion and interpretation of the theoretical results to the paper, and state such interpretation below as well.
>
> For Theorem 3: In appendix D we did give some intuition, which we restate and expand here: Suppose there are in total $K$ blocks, and blocks are i.i.d. We still allow arbitrary dependence within a block. The length of each block is $q$. Theorem 3 relies on the result in Appendix D.3 that DA produces iterates whose error is of order $E[\\| w_t - w^*\\|^2 ]=O(q^2/t)$.
>
> Consider dual averaging with the knowledge of the block structure $q$. Then the rate $1/K = q/t$ can be achieved by executing DA using one randomly chosen data point within a block, throwing away the rest in that same block. Such selection produces $K$ i.i.d. samples from the distribution. In comparison, the rate in $O(q^2/t)$ is worse off by a factor of $q$ due to not knowing the block-structure information.
>
> For Theorem 4: Theorem 4 says in the face of independent data with adversarial corruption, DA produces iterates that converge to the optimal solution at rate $1/t$ up to some error caused by corruption. From this result we can tell when DA retains convergence.
> First, the most general case where our result shows DA convergence is when $o(t)$ data points are corrupted, in which case the corruption per item is $\delta = o(t)/t = o(1)$.
>
> Furthermore, one can get an explicit rate if the corruption per data is smaller than that. For example, if it is of order $\delta = o(1/t)$, then the fast rate $1/t$ is retained.
>
> We will add the above discussion to the main body in a revised version.
>
> > The background of the paper goes almost to page 5, which does not leave enough space to present and discuss the main results in the paper. It may be better to make the background concise and some may be moved to the supplementary material.
> Response:
>
> Response:
>
> We agree that there is quite a bit of background material. First, an additional page of content is allowed for camera-ready papers, which would allow us to add experiments, as well as add the intuition that you requested above.
>
> You also suggested condensing preliminaries. Our preference would be to primarily use the extra page for fitting experiments and the new intuition. Aggressively shortening the preliminaries might make them too dense, and we note that reviewers dWex and 4sGJ found the paper well-written, which probably required close to the current length. We will try to condense it somewhat, though. We expect that condensation to yield perhaps ⅓ of a page.
>
>
> > The abstract mentioned the numerical results of the paper; however, the experiments are not presented, until the end of the supplementary material.
>
> Response:
>
> Reviewer dWex also asked for the experiments to be included in the body. Here is our plan for doing that, quoting from above:
> “In order to incorporate the experiments in the body, we will use the additional page allowed by NeurIPS camera-ready papers to fit those experiments. Additionally, we plan to slightly shorten some of the preliminaries, as you can see in our reply to reviewer nihF below. This will yield even more space for experiments.”

---

### Official Review · Reviewer_dWex · 2022-07-15

**Rating:** 5
**Confidence:** 2
**Ethics Flag:** Yes
**Soundness:** 3 good
**Presentation:** 4 excellent
**Contribution:** 2 fair

**Summary:**

The paper discusses the problem of online fair allocation under nonstationary data distributions. The authors describe the problem of fair item allocation at stake and the PACE algorithm they base their analysis on. Then, they derive several regret bounds under several types of nonstationarity, namely adversarial input, ergodic and periodic distributions. Then, they provide a summary of the proof techniques used to achieve these results.

**Questions:**

In the original study of dual averaging for strongly convex objectives, Xiao et al. provided a method that could be sped-up with a surrogate function $h$, leading to a refined regret bound (at the cost of the introduction of a new hyperparameter $\beta_t$). Would such a function be conceivable in this case to improve the PACE performance?

Is there an intuition as to why a soft allocation will not lead to better rates for the PACE algorithm? Pure allocation does sound necessary in some settings but I wonder why a partial allocation could not be better if the setting allows it. Is it because we only care about expected regret here, and not about the variance?

**Limitations:**

The limitations of the different assumptions are clearly discussed throughout the paper.

**Strengths And Weaknesses:**

The paper is well-written and introduces carefully the setup of the problem. Every assumption is justified and seems reasonable.
The resulting regret bounds are clear and provided with a quick interpretation.

The main issue of the paper comes from the empirical study: although the bounds are interesting, it would have been a preferable sanity-check to have toy experiments evidencing that the dependence in the ergodicity/TV-norm/etc.  reflects the true behavior of the regret. The proposed experiments are not in the main paper (and I believe that some results could be shortened/factorized to leave additional space in the main paper) and even the experiments provided in the supplementary materials are not very convincing. They show that PACE algorithm still holds under some nonstationary settings, but they fail to evidence the impact of increasing corruption/deviation from a stationary distribution, which is exactly what the theoretical results state.

---

> ### Author Response · Authors · 2022-07-29
> **response**
>
> > In the original study of dual averaging for strongly convex objectives, Xiao et al. provided a method that could be sped-up with a surrogate function h, leading to a refined regret bound (at the cost of the introduction of a new hyperparameter βt). Would such a function be conceivable in this case to improve the PACE performance?
>
> Response:
>
> No, this would not lead to an improvement in terms of theory. With or without an extra surrogate function h, the regret achieved by DA in the strongly convex case is $O( \log t)$. The advantage of this update rule is that it admits a particularly simple update rule while enjoying the optimal regret rate among such rules. One could add an identical additional log regularizer $h(\beta) =- \log \beta$ and have a similar update rule; this would yield the same theoretical rate, but more conservative updates in practice, which is likely not helpful.
>
>
> > Is there an intuition as to why a soft allocation will not lead to better rates for the PACE algorithm? Pure allocation does sound necessary in some settings but I wonder why a partial allocation could not be better if the setting allows it. Is it because we only care about expected regret here, and not about the variance?
>
> Response:
>
> We agree that, intuitively, you might think that partial allocation would be better. But in fact, this is not true, and it is because of the asymptotic and stochastic nature of the results. It is easiest to think of this in a setting with a finite set of possible items. Assume stationarity for simplicity. In that case, as T grows large, every item occurs on the order of T times. Thus, as T grows large any discrete allocation can approximate any fractional allocation. Now, with a continuum you have to be more careful because there will be no duplicate items, but the same effect occurs with items that are close to each other being, more or less, substitutable.
>
> Secondly, note that if valuations are continuously distributed, then ties actually occur with probability zero (i.e. $\beta^t_i v_{i} (\theta^t) \ne \beta^t_k v_{k} (\theta^t)$ for all $i,k$), so in general we cannot hope to leverage this fact for improving PACE. We would need to move to a different algorithm that does not allocate to the highest bidder.
>
> In an adversarial setting, as opposed to stochastic, fractional allocation might be useful. For example, suppose all values are zero until the last time step, and then everyone has value 1. In that case we clearly need fractional allocation.

---

> ### Author Response · Authors · 2022-08-02
> **On experiments / "main issue"**
>
> > The main issue of the paper comes from the empirical study: although the bounds are interesting, it would have been a preferable sanity-check to have toy experiments evidencing that the dependence in the ergodicity/TV-norm/etc. reflects the true behavior of the regret. The proposed experiments are not in the main paper (and I believe that some results could be shortened/factorized to leave additional space in the main paper) and even the experiments provided in the supplementary materials are not very convincing. They show that PACE algorithm still holds under some nonstationary settings, but they fail to evidence the impact of increasing corruption/deviation from a stationary distribution, which is exactly what the theoretical results state.
>
> While our theory suggests that the performance guarantees of PACE degrade as the degree of nonstationary grows, that is actually not what we intended to show in those experiments. We view our theoretical results as supporting the conclusion that PACE is likely to work well on many different types of nonstationary input observed in practice; yet real instances will not look exactly like any of the models we studied. We were thus trying to highlight the robustness of PACE: essentially, we want it to work in practice for a broad class of input types. To this end, the plots do show that the allowed types of nonstationarity (e.g., mild but arbitrary corruption, Markov and periodic) does not stop PACE from converging to an equilibrium in the long run. In fact, the observed convergence behavior of PACE under (simulated) nonstationary arrivals is very similar to the case of i.i.d. Inputs. The arrivals are generated using the 3 functions in generate_item_arrivals.py starting with “sample_all_arrivals_”, which are self-explanatory.
> Nonetheless, we do agree that it would be useful to add experiment results that show the practical performance of PACE under settings that correspond more directly to the theoretical models studied. To that end, we have conducted more experiments that show the effect of nonstationarity on the performance of PACE and will add the results to the revised manuscript. For example, below is a plot that compares the convergence of the pacing multipliers that PACE generates, for item arrivals that are i.i.d. (following distribution $s$, blue curve) or follow perturbed distributions (that are close to but different from $s$, two other curves). Here, $\beta_i^*$ are the equilibrium utility prices given item supplies $s$. The error metric is the maximum relative error in $\beta^t_i$. For each instance, we randomly generated 10 sample paths and ran PACE 10 times to obtain the mean values and standard errors of the error metric. The two perturbed instances are generated by choosing item subsets of different sizes and moving all of their probabilities to a single item. $delta$ is as in (8) in the paper, which measures the overall perturbation, and is between 0 and 1.
>
> [anonymized imgur.com url to plot](https://imgur.com/a/XbUGT6h)
>
> In order to incorporate the experiments in the body, we will use the additional page allowed by NeurIPS camera-ready papers to fit those experiments. Additionally, we plan to slightly shorten some of the preliminaries, as you can see in our reply to reviewer nihF below. This will yield even more space for experiments.

---

### Meta-Review · Area_Chair_9Xd1 · 2022-08-26

**Recommendation:** Accept
**Confidence:** Less certain

**Metareview:**

This paper studies the problem of online fair allocation. PACE algorithm has been proposed earlier to tackle this problem. Earlier analysis of this algorithm was under i.i.d. assumption. This paper presents a significant extension of the earlier work, providing guarantees for PACE under a significantly less restrictive data generating processes, e.g. adversarially-corrupted stochastic input, ergodic input, and block-independent (including periodic) input. This extension advances theoretical understanding of PACE algorithm and is of interest to Neurips community. The paper is recommended for acceptance.

**Award:**

No

---

### Decision · Program_Chairs · 2022-09-14

Accept